# Automated Labelling using an Attention model for Radiology reports of MRI scans (ALARM)

**David A. Wood**[1]                                            DAVID.WOOD@KCL.AC.UK
**Jeremy Lynch**[2]                                            JEREMY.LYNCH@NHS.UK
**Sina Kafiabadi**[2]                                            SKAFIABADI@NHS.NET
**Emily Guilhem**[2]                                      EMILY.GUILHEM@DOCTORS.ORG.UK
**Aisha Al Busaidi**[2]                                      AYISHA.ALBUSAIDI@NHS.NET
**Antanas Montvila**[2]                                    MONTVILA.ANTANAS@GMAIL.COM
**Thomas Varsavsky**[1]                                   THOMAS.VARSAVSKY@KCL.AC.UK
**Juveria Siddiqui**[2]                                      JUVERIA.SIDDIQUI1@NHS.NET
**Naveen Gadapa**[2]                                        NAVEEN.GADAPA@NHS.NET
**Matthew Townend**[3]                                   MATTHEW.TOWNEND@WWL.NHS.UK
**Martin Kiik**[1]                                              MARTIN.KIIK@KCL.AC.UK
**Keena Patel**[1]                                            KEENA.PATEL@KCL.AC.UK
**Gareth Barker**[4]                                        GARETH.BARKER@KCL.AC.UK
**Sebastian Ourselin**[1]                                  SEBASTIEN.OURSELIN@KCL.AC.UK
**James H. Cole**[4,5]                                         JAMES.COLE@UCL.AC.UK
**Thomas C. Booth**[1,2]                                      THOMAS.BOOTH@KCL.AC.UK

[1] *School of Biomedical Engineering, King's College London*

[2] *King's College Hospital, London, UK*

[3] *Wrightington, Wigan & Leigh NHSFT*

[4] *Institute of Psychiatry, Psychology & Neuroscience, King's College London*

[5] *Centre for Medical Image Computing, Dementia Research, University College London*

## Abstract

Labelling large datasets for training high-capacity neural networks is a major obstacle to the development of deep learning-based medical imaging applications. Here we present a transformer-based network for magnetic resonance imaging (MRI) radiology report classification which automates this task by assigning image labels on the basis of free-text expert radiology reports. Our model's performance is comparable to that of an expert radiologist, and better than that of an expert physician, demonstrating the feasibility of this approach. We make our code available for researchers to label their own MRI datasets for medical imaging applications.

## 1. Introduction

Deep learning-based computer vision systems hold promise for a variety of medical imaging applications. However, a rate-limiting step is the labelling of large datasets for model training, as archived images are rarely stored in a manner suitable for supervised learning (i.e with categorical labels). Unlike traditional computer vision tasks where image annotation

is simple (e.g labelling dog, cat, bird etc) and solutions exist for large-scale labelling (e.g crowd-sourcing, (Deng et al., 2009)), assigning radiological labels to magnetic resonance imaging (MRI) scans is a complex process requiring considerable domain knowledge and experience. A promising approach to automate this task is to train a natural language processing (NLP) model to derive labels from radiology reports and to assign these labels to the corresponding MRI scans. To date, however, there has been no demonstration of this labelling technique for MRI scans (Pons et al., 2016). We ascribe this to i) the greater lexical complexity of MRI reports compared with that of other imaging modalities such as computed tomography (CT) which reflects the high soft tissue contrast resolution of MRI allowing more detailed description of abnormalities and enabling more refined diagnoses, and ii) the difficulty of training sophisticated language models with relatively small corpora. Following the open-source release of several state-of-the-art transformer models (Vaswani et al., 2017) (Devlin et al., 2018)(Lee et al., 2019)(Alsentzer et al., 2019), however, this latter issue has been somewhat resolved. Instead of training from scratch, domain-specific models can be fine-tuned from these general language models (which are pre-trained on huge collections of unannotated text). Because considerable low-level language comprehension is inherited from these parent models, fewer domain-specifc labelled examples are necessary for training, making previously intractable specialised language tasks feasible.

In this work we introduce a transformer-based neuroradiology report classifier for automatically labelling hospital head MRI datasets. The model is built on top of the recently released BioBERT language model (Alsentzer et al., 2019) which we modify and fine-tune for our report corpus. The model is trained on a small subset of these reports, manually labelled by a team of clinical neuroradiologists into five clinically useful categories that have been carefully designed with the goal of developing an automated triage system. Encouragingly, the classification performance of the model is comparable to that of an experienced neuroradiologist, and better than that of an experienced neurologist and stroke physician, demonstrating feasibility that our model can automate the labelling of large MRI head datasets. We make code available online for researchers to label imaging datasets for their own computer vision applications.

## 2. Related work

To the best of our knowledge, classification of MRI head radiology reports for automated dataset labelling has not been previously reported, although several works have presented models for classifying head (Yadav et al., 2016) and lung (Chen et al., 2017) (Banerjee et al., 2019) computed tomography (CT) reports. These range from simple rule-based and bag of words (BoW) models, to more sophisticated deep learning-based approaches, with developments in general mirroring those of the wider NLP field. The closest work to ours is that of (Shin et al., 2017) and (Zech et al., 2018). (Zech et al., 2018) trained a linear lasso model using a combination of N-gram bag of words (BoW) vectors and average word embeddings pre-trained using word2vec (Mikolov et al., 2013) to detect critical findings in CT head reports. (Shin et al., 2017) presented a CNN model for classifying reports into several categories (normal, abnormal, acute stroke, acute intracranial bleed, acute mass effect, and acute hydrocephalus) again using word2vec embeddings. Although these models

outperformed the previous state-of-the-art, their classification performance (91.4% and 87% average accuracy over all categories for (Zech et al., 2018) and (Shin et al., 2017), respectively) is still below that required to build a clinically useful labelling system.

A fundamental limitation of these models, and of NLP models employing word2vec embeddings generally, is that a given word's representation is context-independent. This issue was addressed sequentially in 2018, beginning with the introduction of deep contextualized word embeddings derived from a bi-directional long-short-term memory (LSTM) language model by (Peters et al., 2018) and culminating in the work of (Devlin et al., 2018), which introduced a bi-directional transformer (BERT) model. BERT was pre-trained on BooksCorpus (800 million words) (Zhu et al., 2015) and English Wikipedia (2.5 billion words), and achieved state-of-the-art performance on a number of NLP tasks. Following the open-source release of the pre-trained BERT model and code, a number of researchers released domain-specific versions of BERT, including BioBERT (Lee et al., 2019), trained on PubMed abstracts (4.5 billion words) and PMC full-text articles (13.5 billion words), and ClinicalBERT (Alsentzer et al., 2019), trained on 2 million clinical notes from the MIMIC-III v1.4 database. These fine-tuned models were shown to outperform the original BERT model on NLP tasks in these more specialised domains.

Our work integrates and builds on that of (Zech et al., 2018), (Shin et al., 2017), (Devlin et al., 2018), and (Lee et al., 2019), introducing for the first time a transformer-based model, built on top of BioBERT, to classify MRI head radiology reports to enable automated labelling of large-scale retrospective hospital datasets. We also introduce a custom attention model to aggregate the encoder output, rather than use the classification token ('CLS') as suggested in the original BERT paper, and show that this leads to improved performance and interpretability, outperforming simpler fixed embedding and word2vec-based models.

## 3. Methods

### 3.1. Data

The UK's National Health Research Authority and Research Ethics Committee approved this study. 126,556 radiology reports produced by expert neuroradiologists (UK consultant grade), consisting of all adult ($> 18$ years old) MRI head examinations performed at Kings College Hospital, London, UK (KCH) between 2008 and 2019, were used in this study. The reports were extracted from the Computerised Radiology Information System (CRIS) (Healthcare Software Systems, Mansfield, UK) and all data was de-identified. Unlike (Zech et al., 2018) and (Shin et al., 2017), the reports are largely unstructured, with more than 75% of reports missing a clearly defined conclusion section, and over 25% having no well defined clinical information in the referring doctor's section (this section should contain the clinical features that the patient presented with, e.g. right-sided weakness, or background clinical history, e.g. previous stroke). Each report is typically composed of 5-10 sentences of image interpretation, and sometimes included information from the scan protocol, comments regarding the patient's clinical history, and recommended actions for the referring doctor. The report had often been transcribed using voice recognition software (Dragon, Burlington, US).

### 3.1.1. Report annotation

A total of 3000 reports were randomly selected for labelling by a team of neuroradiologists to generate reference standard labels. 2000 reports were independently labelled by two neuroradiologists for the presence or absence of any abnormality, defined on the basis of pre-determined criteria optimized after six months of labelling experiments. We refer to this as the dataset with coarse labels (i.e normal vs. abnormal). Initial agreement between these two labellers was 94.9%, with consensus classification decision made with a third neuroradiologist where there was disagreement. Example reports for both categories appear in appendix A. In addition to this 'coarse dataset', a further 1000 reports were labelled for the presence or absence of each of five more specialised categories of abnormality (vascular abnormality e.g. aneurysm, damage e.g previous brain injury, Fazekas small vessel disease score (Fazekas et al., 1987), mass e.g tumour, and acute stroke), described further in appendix B. We refer to this as the 'granular dataset'. There was unanimous agreement between these three labellers across each category for 95.3% of reports, with a consensus classification decision made with all three neuroradiologists where there was disagreement.

### 3.1.2. Data pre-processing

To ensure our approach was as general as possible, we performed only those pre-processing steps required for compatibility with BERT-based models. Each report was split into a list of integer identifiers, with each identifier uniquely mapped to a token consistent with the vocabulary dictionary used for both BERT and BioBERT training. In addition, the classification ('CLS') token, as described in (Devlin et al., 2018), was prepended to this list, which is then passed as input to our report classifier (described next).

## 3.2. Attention-based BioBERT classifier

Our custom attention-weighted classifier is built on top of the pre-trained BioBERT model. BioBERT takes as input a report in the form of a list of token ids, outputting a 768-dimensional contextualised embedding vector for each token in the report. BioBERT is trained on two semi-supervised tasks: predicting the identity of randomly masked tokens, and predicting whether a given random sentence is likely to directly follow the current one. In the original paper, the authors suggest that down-stream classification tasks should be performed using the 'CLS' token, which can be fine-tuned to provide a good representation of a more specialised domain-specific language task, in this case neuroradiology report labelling. We found in our experiments that the addition of a custom attention module to weight individual word embeddings led to improved performance:

$$c = \sum_t \alpha_t h_t. \tag{1}$$

Here $c$ is the report representation, $\alpha_t$ is a learned context-dependent weighting, and $h_t$ is the embedding for the $t^{\text{th}}$ word. Following (Bahdanau et al., 2014) and (Yang et al., 2016),

we compute $\alpha_t$ using an attention module:

$$\alpha_t = \frac{\exp(h_t^T u)}{\sum_t \exp(h_t u)}$$
$$u_t = \tanh(W h_t + b), \tag{2}$$

where $W$ is a 768 x 768 matrix, and $u$ a 768-dimensional vector - both $W$ and $u$ are learnt during training. In addition to improving performance, the attention mechanism provides a form of interpretability for the model, which allowed us to investigate the reason for classification errors (see later).

The report representation calculated by (1) and (2) was then passed to a 3 layer fully-connected neural network, with ReLU activation and batch-normalization in the intermediate layers, which reduces the dimensionality from 768 to 1 (768 - 512 - 256 - 1). A sigmoid activation then mapped the output to [0,1], representing the probability that a given abnormality is present. The complete model (BioBERT encoder, custom attention module, and classification network) was then trained by minimizing the binary cross-entropy loss between the empirical and predicted label distributions.

### 3.2.1. MODEL BASELINES

Given the absence of a dedicated MRI head report classifier in the literature, we compared our model with the state-of-the-art for CT head report classification, as presented in (Zech et al., 2018). This model is a logistic regression network taking as input the average word2vec embedding for a given report. We followed the preprocessing steps outlined in (Zech et al., 2018), and trained the model on our reference standard labelled MRI reports. We also trained two additional BioBERT classifies, one with contextualised embeddings fixed to that of the pre-trained model (i.e no fine-tuning was performed), and one which uses the 'CLS' token rather than an attention weighted combination of embeddings. We refer to these models as 'FrozenBioBERT' and 'BaseBioBERT', respectively.

## 4. Experiments

We split our coarse and granular datasets into training/validation/test sets of sizes (1500, 200, 300) and (700, 100, 200), respectively. Because this is real-world hospital data, multiple examinations for patients are common and reports describing these separate visits are likely to be highly correlated. To avoid this form of data leakage we performed the split at the level of patients so that no patient appearing in the training set appeared in the validation or test set. The BioBERT encoder component of each model was initialised with the pre-trained parameter values, the attention module context vector was initialised from a zero-centred normal distribution with variance $\sigma = 0.05$, and the whole network was trained for 7 epochs using ADAM (Kingma and Ba, 2014) with initial learning rate 1e-5, decayed by 0.97 after each epoch, on a single NVIDIA GTX 2080ti 11 GB GPU. Final model selection was made on the basis of validation loss, with the best performing model used for evaluation on the test set. The baseline network was trained using code made available by (Zech et al., 2018). Table 1 presents the performance of our model for the 'coarse' binary classification task,

along with that of the baseline models. Also included is the classification performance of a hospital doctor with ten years experience as a stroke physician and neurologist who was trained by our team of neuroradiologists over a six month period to label these reports. Our model achieves very high accuracy (99.4%) sensitivity (99.1%), and specificity (99.6%) on this binary task, outperforming both baseline models, as well as the expert physician.

| Model | accuracy (%) | sensitivity(%) | specificity(%) |
|---|---|---|---|
| Our model | **99.4** | **99.1** | **99.6** |
| base-BioBERT | 97.7 | 97.3 | 97.9 |
| frozen-BioBERT | 96.5 | 96.4 | 96.6 |
| word2vec (Zech et al.) | 91.5 | 79.2 | 97.1 |
| Expert physician | 92.7 | 77.2 | 98.9 |

Table 1: Model and baseline performance on the binary classification task which was to determine the presence or absence of any abnormality in the report. Best performance in bold.

Table 2 presents the performance of our model across the granular abnormality categories, along with that of the baseline models. Also included is the classification performance of a fourth neuroradiologist who had also undergone labelling training, but was blinded to the reference standard labelling process. Averaged over all categories, our model's accuracy (96.7%) was only 1.1% lower than that of the neuroradiologist when classifying the same test set reports. When averaged over all categories, our model (95.2%) was more sensitive than the neuroradiologist (90.2%), outperforming the neuroradiologist in three of the five categories (mass, vascular, and Fazekas).

| Model | damage | | | vascular | | | mass | | | acute stroke | | | Fazekas | | |
|---|---|---|---|---|---|---|---|---|---|---|---|---|---|---|---|
| | acc | sens. | spec. | acc | sens. | spec. | acc | sens. | spec. | acc | sens. | spec. | acc | sens. | spec. |
| Our model | 93.8 | 92.6 | 94.3 | 95.8 | **96.1** | 95.7 | 95.8 | **92.6** | 96.4 | 98.8 | 94.5 | 100 | **99.4** | **100** | 99.3 |
| frozen-BioBERT | 86.3 | 68.5 | 95.2 | 88.5 | 57.7 | 94.3 | 92.3 | 59.3 | 98.6 | 88.7 | 52.7 | 96.2 | 97.1 | 95.5 | 98.2 |
| Neuroradiologist | **96.8** | **96.2** | **97.1** | **96.9** | 84.6 | **99.3** | **96.4** | 77 | **100** | **99.4** | **97.2** | **100** | 99.3 | 96.1 | **100** |

Table 2: Model and baseline performance on the granular classification tasks.

### 4.1. Model analysis

The results from the previous section are more impressive when the difficulty of the classification task is considered, as evidenced by (i) the relatively poor performance of an expert physician trained in report labelling, and (ii) the lexical variation across both categories of reports. As shown in Appendix A, 'normal' reports range from succinct statements such as 'normal study', to the description of a number of findings deemed clinically insignificant or demonstrating normal variation (e.g age appropriate volume loss). Also common are descriptions, preceded by distant negation, of abnormalities which are not present, including those that are being searched for in light of the clinical information (e.g. 'no features sug-

gestive of acute stroke'). The abnormal reports are similarly heterogeneous, ranging from straightforward 'there is a ___ in the ___' statements, to reports which contain no particular description of an abnormality, referring instead to stable appearances of a pre-existing issue (e.g. 'stable disease' when referring to follow-up interval imaging of a brain tumour).

From Tables 1 and 2, it is clear that fine-tuning the pre-trained BioBERT embeddings on our corpus led to considerable improvements in classification performance. The reason for this is explored in Figure 1. Although the pre-trained BioBERT embeddings largely separate the two classes, the boundary is highly non-linear, with many normal and abnormal reports sharing almost identical embeddings. The situation was similar for the word2vec embeddings, with category-dependent clustering broadly evident. Because the word2vec model of (Zech et al., 2018) is a linear classifier, however, the performance of this model was even lower than that of 'frozen-BioBERT' model which can learn non-linear decision boundaries. Clearly, the best representation was achieved by fine-tuning the contextualized embeddings on the classification task. In this case the two groups were widely separated, allowing accurate and confident predictions.

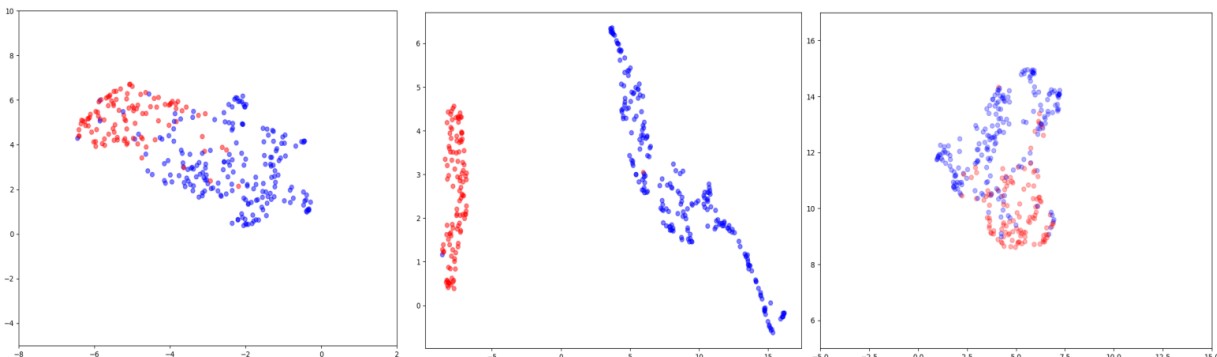

Figure 1: Two-dimensional t-SNE representations of the contextualized embeddings for all reports in the test set before (left) and after (middle) fine-tuning, as well as the average word2vec embedding (right) used by (Zech et al., 2018).

Also crucial to our model's performance is our introduction of a word-level attention layer to weight the relative importance of each word in a given report. More than improving performance, however, this feature offers insight into the model's decision process. As an example, our fine-tuned model only makes two errors in the 'coarse' test set - one false negative and one false positive classification error. We can explore the reason for these model predictions by visualising the word-level attention weights for reports. In the case of the false negative prediction, our model assigned considerable importance to the expressions 'no cortical abnormalities' and 'normal intracranial appearances'. Indeed, this second phrase is a good working definition of our normal category. However, this report was labelled abnormal on account of the presence of a solitary microhaemorrhage. This example highlights a case where the neuroradiologist who reported the original scan reasonably deemed a finding

Figure 2: Visualization of the word-level attention attention weights for two misclassified reports.

insignificant - and used language accordingly - whereas our reference standard labelling team, in order to be as sensitive as possible, pre-determined that any microhaemorrhage is abnormal.

The false positive error was also the result of a difficult case as the report combined a head and spine examination. Because we extracted all head examinations from a real-world hospital CRIS, occasionally the neuroradiologist who reported the original scan decided to include a spine report in the section dedicated for head reports (as opposed to a section for the spine). Such combined reports were very rare as the number of head and spine examinations occurring at the same time was small (less than 0.5% of our data). This particular report described a patient presenting with no intracranial abnormalities, despite having multiple spinal abnormalities. Examining the word-level attention weights for this example, it is clear that our model was erroneously predicting this report to be abnormal on account of these spinal abnormalities, paying particular attention to the 'thinning' of the spinal cord and 'narrowing' of the spinal canal. Clearly, our classification results could be improved even further by removing the 0.5% of examinations where the head and spine examinations were performed concurrently before training. However, our concept had always been to use all written head reports on an 'intention-to-report' basis; we therefore wanted our algorithm to be as generalisable as possible by using all real world data and so elected to use all head reports from a real-world hospital CRIS.

## 4.2. Semi-supervised labelling

Although our granular classifiers performed very well, they are a work in progress and additional labeling of reports by our clinical team is ongoing to increase training set sizes. We note, however, that our 'coarse' classifier allows a semi-supervised form of image labelling

for sub-categories of abnormalities for which no labelled data is available. By examining t-SNE visualisations of abnormal report embeddings, local clustering is seen. Upon closer examination, reports in a given cluster are all found to describe some common pathology. To exploit this we created a web-based annotation tool which enabled us to 'lasso' clusters of data points with a computer mouse and write group labels for these examinations. We make this tool available at https://github.com/tomvars/sifter. Figure 3 shows a binary glioblastoma vs normal dataset generated in this way, along with several example reports. In a similar manner, we have created large specialized datasets for patients with excessive volume loss, recurrent aneurysms, and Alzheimer's disease suitable for computer vision applications.

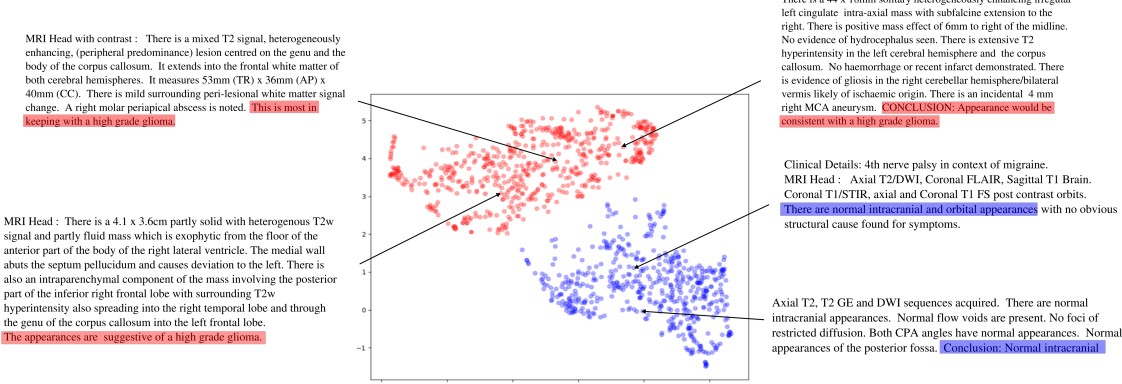

Figure 3: Visualization of the report embeddings for normal (blue) and glioma (red) cases captured using our 'lasso' tool.

## 5. Discussion

By inheriting a high-level understanding of biomedical language from the BioBERT pre-training, our network has achieved expert-level performance on radiology report classification from relatively few labelled examples. Because these reports detail findings made by experienced neuroradiologists, they contain sufficient information to allow the assignment of reference-standard categorical labels to archived MRI images. As such, our model can automatically perform this process in a fraction of the time it takes to perform manually. In this way, we were able to predict and write labels for over 120,000 head MRI scans in under half an hour. Importantly, in the coarse binary case, this automatic labelling can be performed with almost no cost in accuracy and sensitivity compared with manual annotation by a team of neuroradiologists trained for six months to label reports.

## 6. Conclusion

In this work we have presented for the first time a dedicated MRI neuroradiology report classifier, by modifying and fine-tuning the state-of-the-art BioBERT language model. The classification performance of our model, Automated Labelling using an Attention model for Radiology reports (ALARM), was only marginally inferior to an experienced neuroradiologist for granular classification, and better than that of an experienced neurologist and stroke physician, suggesting that ALARM can feasibly be used to automate the labelling of large retrospective hospital MRI head datasets for use with deep learning-based computer vision applications.

## Acknowledgments

This work was supported by The Royal College of Radiologists, King's College Hospital Research and Innovation, King's Health Partners Challenge Fund and the Wellcome/Engineering and Physical Sciences Research Council Center for Medical Engineering (WT 203148/Z/16/Z). We also thank Joe Harper, Justin Sutton, Mark Allin and Sean Hannah at KCH for their informatics AND IT support, Ann-Marie Murtagh at KHP for research process support, and KCL administrative support, particularly from Denise Barton and Patrick Wong.

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

## Appendix A.  Example reports

**Normal study**

MRI BRAIN There is minor generalised prominence of sulci and ventricles. There is no lobar predominance to this minor generalised volume loss. There are scattered punctate patches of high signal on T2/FLAIR within the white matter of both cerebral hemispheres. These number approximately 12. They have a non specific appearance. CONCLUSION: There is minor generalised volume loss without lobar predominance. The extent of this volume loss is probably within normal limits for a patient of this age. Scattered punctate foci of high signal within the white matter of both cerebral hemispheres have a non specific appearance. They are likely to be on the basis of small vessel ischaemic change.

**Normal study**

MRI Head : Ax T2W, Sag T1W volume and Ax FLAIR were obtained. There is a solitary non specific tiny T2 hyperintense focus within the right centrum semiovale, a non specific finding of doubtful significance. The intracranial appearances are otherwise normal.

**Normal study**

MRI Head : MRA images demonstrate loss of signal of the left intracranial ICA. However, normal flow void is seen on the conventional sequences ad this appearance is artifactual. There is otherwise normal appearance of the intracranial arteries. There are normal intracranial appearances with no evidence of vascular malformations or areas of signal abnormality. There are no areas of restricted diffusion or susceptibility artefact. Note is made of prominent arachnoid granulation within the posterior fossa, predominantly right medial transverse sinus.

**Normal study**

Clinical History : First episode of psychosis. MRI Head : Normal intracranial appearances.

**Normal study**

Clinical Details: L sided headache, weakness, numbness. initially aphasic but no receptive deficit, now verbalising as normal. no migraine hx, no PMHx. OE subtle drift L side and reduced sensation, no visual deficit, no speech deficit now Specific question to be answered: rule out stroke MRI Head : Correlation is made to the CT of the same date. No focus of restricted diffusion is demonstrated to indicate an acute infarct. No mass or other focal parenchymal abnormality is identified. The ventricles are of normal size and configuration. The major intracranial vessels demonstrate normal flow related voids. Conclusion: Normal intracranial appearances.

> **Abnormal study**
>
> MR HEAD Axial T2, coronal T1 pre and post gadolinium, axial and sagittal T1 post gadolinium images. Comparison is made with the previous study dated 16.10.08. There are stable appearances to the meningioma within the posterior fossa, arising from the tentorial dura overlying the vermis and supero-medial aspects of the cerebellar hemispheres. No further intracranial abnormalities are shown.

> **Abnormal study**
>
> Technique: Axial T2, Coronal Pre- and Post-Gad T1, Axial Post-Gad, DWI Findings: Comparison has been made with the previous MRI brain scan dated 09/10/2008. Stable intracranial appearances. The enhancing scar tissue at the site of the previous craniotomy is unchanged in appearance. Impression: No evidence of disease recurrence.

> **Abnormal study**
>
> MRI BRAIN Axial T2, axial FLAIR, coronal T1 pre and post contrast, axial T1 post contrast. There are multiple, cortical and subcortical, T2/FLAIR hyperintense lesions in both hemispheres. There is associated restricted diffusion and extensive, subtle, nodular enhancement of the cortex and overlying meninges particularly in the right Rolandic sulcus. CONCLUSION The appearances are most in keeping with extensive meningeal metastatic disease and underlying, multifocal, cortical infarction.

## Appendix B. Granular classification definitions

The neuroradiology label classification was refined after 6 neuroradiology meetings, each following a practice classification task of 100 reports. These definitions are for intracranial findings.

Note that a separate filter also exists in terms of a coarse classifier for "normal" OR "abnormal" so abnormalities not included here will still be flagged up.

### B.1. Fazekas

(Fazekas et al., 1987) gives a classification system for white matter lesions (WMLs):

1. Mild - punctate WMLS: Fazekas I

2. Moderate - confluent WMLs: Fazekas II

3. Severe - extensive confluent WMLs: Fazekas III

To create a binary categorical variable from this system, if the report was unsure/normal or mild this would be categorized as '0' as this never requires treatment for cardiovascular risk factors. However, if there is a description of moderate or severe WMLs, the report

would be categorized as "1" as these cases sometimes require treatment for cardiovascular risk factors.

## B.2. Mass

We have attempted to emulate the decision making of a neuroradiologist for all categories. Note that a finding that might generate a referral to a multidisciplinary meeting for clarification would be included within the granular category e.g. an arachnoid cyst may be ignored in clinical practice, but we included it in the 'mass' granular category as these are sometimes referred to a multidisciplinary meeting for expert review by non-experts. Thus our classification is sensitive to ensure patient safety.

All the following are categorized as '1' (mass):

- Neoplasms

  - infiltrative tumors
  - extra-axial masses e.g. vestibular schwannoma
  - tumor debulking or partial resection as this includes cavity plus tumor (labelled as both "damage" and "mass")
  - pituitary adenomas
  - ependymal / subependymal / local meningeal enhancement in the context of a history of an aggressive infiltrative tumor

- Abscess

- Cysts

  - retrocebellar cyst is included but mega cisterna magna is ignored
  - pineal cysts and choroid fissure cysts
  - Including Rathke's cleft cysts

- Focal cortical dysplasia, nodular grey matter heterotopia, subependymal nodules and subcortical tubers

- Lipoma

- Chronic subdural haematoma / hygroma (i.e. CSF equivalent)

- Ignore perivascular spaces unless giant

## B.3. Vascular

Note that a finding that might generate a referral to a multidisciplinary meeting for clarification would be included within the category e.g. developmental venous anomaly may be ignored in clinical practice, but we included it in the "vascular" granular category.

- Aneurysms

- including coiled aneurysms regardless of whether there is a residual neck or not

- Arteriovenous malformation

- Arteriovenous dural fistula

- Cavernoma

- Capillary telangiectasia

- Old / non-specific microhemorrhages

- Petechial hemorrhage

- Developmental venous anomaly

- Venous sinus thrombosis

  - in cases of sluggish flow – if strong suspicion of thrombus include otherwise ignore

- Arterial occlusion / flow void abnormality or absence

- Venous sinus tumor invasion (labelled as both "vascular" and "mass")

- Arterial stenosis / attenuation – Include if abnormal. If constitutional / normal variant ignore.

## B.4. damage

- Gliosis

- Encephalomalacia

- Cavity

- If the patient has had a craniotomy or biopsy there is likely damage – however, for example in the case of a burr-hole and drain previously inserted into the extra-axial space, this does not automatically constitute damage

- "Post-operative changes / appearances" include as damage

- Tumor debulking or partial resection as this includes cavity plus tumour (labelled as both "damage" and "mass")

- Chronic infarct / sequelae of infarct

- Chronic haemorrhage / sequelae of haemorrhage (with / without hemosiderin staining)

- Cortical laminar necrosis

### B.5. Stroke

– Acute / subacute infarct (if demonstrating restricted diffusion)

    – Include if there are other descriptors indicating a subacute nature such as swelling or "maturing infarct" even though restricted diffusion has normalised

– Parenchymal post-operative restricted diffusion / retraction injury (labelled as both 'damage' and 'stroke')

– Chronic infarct / sequelae of infarct should be labelled under 'damage'

### B.6. Granular category distribution

| Category | abnormal (coarse) | damage | vascular | mass | acute stroke | Fazekas |
|---|---|---|---|---|---|---|
| Num. positive examples | 698 | 336 | 170 | 171 | 156 | 151 |

Table 3: Number of positive examples for each category.

