# OpenReview forum: "Automated Labelling using an Attention model for Radiology reports of MRI scans (ALARM)"
_MIDL.io/2020/Conference — MIDL 2020_

### Official Review · AnonReviewer2 · 2020-03-05
**Relevant contribution to an increasingly pressing area**

**Rating:** 3
**Confidence:** 4
**Recommendation:** Poster

**Summary:**

The authors propose to and show how to turn a state-of-the-art NLP model, BioBERT, into a tool to solve a basic, but relevant text classification task for free-text radiology reports written (dictated) for head MRI exams. To this end, from a large collection of reports, a total of 3.000 reports were expert-curated into 2 classes for 2/3 of the cases, and into a multi-hot vector for five classes for the remaining 1.000 cases.

The results show an improvement over previous research, competitive with or outperforming a trained human observer on a limited set of test cases. The authors use BioBERT, a NLP tool based on BERT, to turn a report into a richer representation that can be run through a word-level attention mechanism to inspect the words BioBERT attended to most. The result of this attention module is then run through a dense NN for classification, which was also trained by the authors.

The paper shows that limited efforts and modest hardware is sufficient to yield a valuable text classifier that even allows a certain level of interpretability, and that can be used to quickly crawl large report databases.

**Strengths:**

The authors justify why they deviate from the proposed way of fine-tuning BERT-based models for classification, by reporting improved performance if not the result on the [CLS] token alone is used for classification, but the full embedded report representation, run through another self-trained attention module. This is a thought that seems to be justified by the success, even though no theoretical explanation or numerical validation is given.

The paper is clearly structured, consistent in the writing, and sound in its methodology. It presents a useful development, and convincing results.
The evaluation against the closest existing tools (though they are not based on a comparable technology) is augmented by a comparison with a trained human observer, which is not often explicitly done in this research.
The promised release of a text data analysis tool based on tSNE adds to the practical usefulness of the work.


**Weaknesses:**

The for me most significant lack is a clear description how
* the ground truth was established;
* the human observer performance seen in the comparisons was assessed against this GT. I would have assumed that there is no performance difference between the ground truth (established by trained human observers, after all) and the rating of another long-trained (as the authors point out) human observer. Because there _is_ a strong difference, there must be a reason why, which I would really like to know.

The further comments may serve as suggestions for further work. Perhaps it might even be possible to include an implementation of the first point before final submission, if the authors agree.

The methodological contribution (word-level attention visualization) is not very strong, as BERT by itself is built to facilitate this type of introspection, and it has been used in many subsequent works. Also, the attention module's output is input to the 3-layer classifier network that does the actual "judgement", but is not explained or utilized in the explanation (or e.g. used to derive decision uncertainty, which would be a very simple addition).

Also, in my opinion in particular the explanations shown in the two false negative/false positive cases show the major difficulty with some types of explainability mechanisms like the one presented: they might help to elucidate WHY a DNN was wrong, once you know it was, but it does not help to assess IF it was wrong. To achieve this, one way might be to assign not only attention to words, but also a certainty metric, so that the network can be trained to be less certain when it is wrong (compare e.g. Mukhoti/Gal 2018).

**Detailed Comments:**

* In Sec 4. l. 4, change "preformed" --> "performed"
* There is a textual reference to Tables 1 and 2, where the word "Tables" is missing (Sec. 4.1)
* The GitHub link does not exist yet, but it gives away the origin of the contribution (breaks anonymisation)

**Justification Of Rating:**

A slight lack of justification for the particular setup with a new attention module and subsequent classifier network and the unexplained strong difference between human observer and ground truth make me hope that a "weak accept" encourages the authors to improve the submission.
In this case, and if the model and training setup as well as the data annotation tool will indeed be released, I can imagine that the interest of the community might be high enough to warrant an upgrade to an oral presentation.

**Paper Type:**

validation/application paper

**Questions To Address In The Rebuttal:**

See the points in the "weaknesses" section above, please!

**Special Issue:**

no

---

> ### Author Response · Authors · 2020-03-27
> **Thank you for the comments, please see our reply below**
>
> RESPONSE 1 We thank the reviewer for their comments. Regarding how the ground truth labels were established, we thank the reviewer for asking for clarification. We have made this clearer in the manuscript, adding a detailed description of how the categories are defined in the appendix (the neuroradiology granular label classification rules).
>
> - Regarding why there exists a performance difference between the ground truth (established by human observers) and that of a trained human observer, in 3.1.1 we indicate that the ground truth was generated by three neuroradiologists. Because there was initial unanimous agreement of 95.3%, 4.7% of reports had to be labelled following a consensus with all three neuroradiologists to give the ground truth (we refer to this as the ‘reference standard’ in the manuscript). Thus, the ground truth combines the expertise of several neuroradiologists. Conversely, the human observer (a fourth neuroradiologist), although trained in the same way, did not confer with these three neuroradiologists when classifying reports. As such, the performance was marginally inferior. (please also see response to reviewer 1 for discussion about the use of a stroke doctor/neurologist
>
> - Whilst this hopefully answers the question of why the performance is different, another question might be “why did we chose to compare the algorithm to a single blinded observer in addition, rather than against the ground truth alone?” It is well known in medical data classification tasks that the rate limiting step is typically labelling by clinicians. We wanted to determine whether a single experienced neuroradiologist would be suitable for such a task, rather than using a labour-intensive consensus of three experienced neuroradiologists.
>
> - Furthermore, our approach to a specific classification is similar to that used recently where the classification from a group of radiologists was the ‘reference standard’ and comparison was made to an individual radiologist, and therefore appears to be a sensible strategy (Ardila, D., Kiraly, A.P., Bharadwaj, S. et al. End-to-end lung cancer screening with three-dimensional deep learning on low-dose chest computed tomography. Nat Med 25, 954–961 (2019).). However, we even improved the previous approach of determining the ‘reference standard’ for our task by using the consensus results of three radiologists rather than the ‘average’ result.
>
> - RESPONSE 2 We also thank the reviewer for highlighting some typographical errors – we have changed the manuscript to correct these. Thank you.
>
> - RESPONSE 3 Regarding the reviewer's point that 'the attention module's output is input to the 3-layer classifier network that does the actual "judgement", but is not explained or utilized in the explanation', we thank the reviewer for this comment. We take the reviewer's point that attention visualizations are an imperfect  form of model explainability; our inclusion of a custom attention module is ultimately a result of experiments which demonstrate improved performance. On that note we also thank the reviewer for highlighting that we don't explicitly state the performance of baselines using the 'CLS' token, or a simple average word embedding. We have now updated the manuscript to include the results using these baselines - the attention module improves accuracy by around 3% on the course classification task. Following an influential work (Yang et al. 2016, Hierarchical attention networks for document classification), we simply visualize the attention weights to confirm that qualitatively informative words are being used by the model for classification to help shed some light on model behavior. Ultimately, this paper is about accurate labeling of large scale imaging datasets which is a crucial bottleneck rather than examining DNN decisions.
>
> We also thank the reviewer for suggesting an avenue for further work, namely that our model might benefit from the inclusion of certainty metric like that in Mukhoti/Gal 2018. We were not aware of this interesting work and will try to pursue this in future.
>
> We hope that our description of why there exists a  difference between the human observer performance and the ground truth, as well as our intention to make all our code and the tSNE annotation tool available to other researchers upon acceptance so that they can label their own large hospital datasets will mean that the paper will warrant an upgrade to oral as the reviewer kindly suggested. We thank the reviewer again for their comments which have help improved our submission.

---

### Official Review · AnonReviewer1 · 2020-03-11
**Highly relevant, although not directly on imaging data**

**Rating:** 4
**Confidence:** 4
**Recommendation:** Oral

**Summary:**

This work presents a method for automated labeling of radiology report. The method used a standard pretrained classifier (BioBERT) which is extended by transformer-based model and a custom attention function. The task is split into two subtasks: binary and granular classification. Method's restuls have significantly improved over reference methods and experts.



**Strengths:**

This work is very important, for training automated medical image classifiers in the future without the need for manually labeling large datasets.
The dataset for this work is very useful and authors put a lot of effort into labelling these reports. The validation of the work is well done, results are convincing.
Paper is well-written and structured.
Examples of results are a valuable addition.

**Weaknesses:**

Although this work is not using imaging data, this is a future application of the method.
It is not completely clear how the granular classification tasks are defined. For example Fazekas is a score system, is the classifier Fazekas normal yes/no, or predicting the exact score?
No further weaknesses.

**Justification Of Rating:**

Although this work is not using imaging data, is has a very strong connection to it and therefore I find this work highly relevant for MIDL. The validation of the work is well done, results are convincing.

**Paper Type:**

both

**Special Issue:**

no

---

> ### Author Response · Authors · 2020-03-27
> **thank you for the comments, please see our reply below**
>
> RESPONSE 1 We thank the reviewer for their comments. Regarding the fact that our work isn’t directly on imaging data, we believe that it directly pertains to a critical bottleneck in the ‘deep learning for image analysis’ pipeline, namely the difficulty obtaining large labelled datasets of medical images, and we thank the reviewer for highlighting the connection we are emphasising.
>
> - Regarding the definition of the granular classification categories, we thank the reviewer for asking for clarification here. We have added the neuroradiology granular label classification rules as an appendix to show how the granular classification tasks are defined. This neuroradiology label classification was refined after 6 neuroradiology meetings, each following a practice classification task of 100 reports. We have attempted to emulate the decision making of a neuroradiologist. Note that a finding that might generate a referral to a multidisciplinary meeting for clarification would be included within that category e.g. an arachnoid cyst may be ignored in clinical practice, but we included it in the “mass” granular category. Thus our classification is sensitive to ensure patient safety.
>
> - For the example of Fazekas that you mention, these white matter changes are classified according to Fazekas (as referenced in section 3.1.1 F Fazekas, John Chawluk, Abass Alavi, H.I. Hurtig, and R.A. Zimmerman. Mr signal abnormalities at 1.5 t in alzheimer’s dementia and normal aging. AJR. American journal of roentgenology, 149:351–6, 08 1987. doi: 10.2214/ajr.149.2.351. ):
>
> 1. Mild - punctate WMLs: Fazekas I
>
> 2. Moderate - confluent WMLs: Fazekas II
>
> 3. Severe - extensive confluent WMLs: Fazekas III
>
> - To create a binary categorical variable from this system, if the report was unsure/normal or mild this would NOT be categorized as “0” as this never requires treatment for cardiovascular risk factors. However, if it described moderate or severe WMLs this would be categorized as “”1” as these cases sometimes require treatment for cardiovascular risk factors.

---

### Official Review · AnonReviewer3 · 2020-03-11
**Interesting application paper on automated labelling of free-text radiology reports**

**Rating:** 4
**Confidence:** 4
**Recommendation:** Oral

**Summary:**

The paper proposes a method to automatically classify free-text radiology reports. The algorithm is built on top of a pretrained BioBERT model that converts text terms ("tokens") to high-dimensional representations. As a novelty in this work, an attention module is used to compute a weighted average of the high-dimensional representations of all tokens in the report. This average representation is passed to a 3-layer fully connected network to predict a label. The entire network is trained end-to-end. Several experiments are done on a dataset of 3000 labelled reports. The model is compared to a simplified version (with fixed pre-trained weights of the BioBERT model), to an existing approach word2vec, and to humans. The obtained prediction accuracies are impressive.

**Strengths:**

- Clearly written manuscript.
- Relevant and interesting application.
- Well-designed and carefully performed evaluation experiments.
- Results are good.
- Thanks to the attention module, the network is quite interpretable.


**Weaknesses:**

- The experiments did not evaluate the effect of adding the custom attention module on the performance. They only report in the Method section (3.2) that it led to improved performance, but no results are shown to confirm this.
- The class sizes for the labelled data are not reported.

**Detailed Comments:**

- From section 3.1.2, it seems that each report is considered as a single sentence. ("Each report was split into a list of integer identifiers....... sentence separation ('SEP') tokens,.., are appended to the beginning and end of this list"). Do I interpret this correctly? If yes, please clarify why you didn't split the report in sentences. If no, please rephrase.
- Please list the class sizes for the labelled data.

**Justification Of Rating:**

This paper presents an interesting and original application, and shows very promising results on a large dataset. The method seems well-designed, and has some incremental novelty. This is a good application paper.

**Paper Type:**

validation/application paper

**Questions To Address In The Rebuttal:**

- Please compare the performance with and without the custom attention module and show the results.

**Special Issue:**

yes

---

> ### Author Response · Authors · 2020-03-27
> **Thank you for the comments, please see our reply**
>
> RESPONSE 1 We thank the reviewer for their comments. Regarding the performance of our model without the custom attention module, we thank the reviewer for asking for clarification about this. We have updated the manuscript with results using the CLS token – the suggested technique form the original BERT paper - as well as with a simple average of contextualized word embeddings i.e. the attention weights for each word are equal an given by 1/length_sentence. Our attention network outperforms both by ~3%.
>
> - RESPONSE 2 Regarding the use of the ‘SEP’ token, thank you for asking for clarification about this. The reviewer is correct – the SEP token was used to separate sentences - we have rephrased the manuscript to make this clearer.
>
> - RESPONSE 3 Regarding the class sizes for the labelled data, we thank the reviewer for highlighting this. We have now included these is the manuscript.

---

> > ### Comment · AnonReviewer3 · 2020-04-03
> > **Thank you for clarification**
> >
> > I thank the authors for the clarification. No change to initial rating.

---

### Official Review · AnonReviewer4 · 2020-03-18
**Very interesting application of NLP to classification of neuroradiology reports with impressive experiments**

**Rating:** 4
**Confidence:** 4
**Recommendation:** Best Paper Award, Oral

**Summary:**

key ideas: In this work the authors present a very interesting application of NLP to classification of neuroradiology reports. Their contribution is to modify and re-tune the state-of-the-art BioBERT language model for the task of classifying radiological descriptions of historical MRI head scans into normal and abnormal as well as several subcategories . The
classication performance of the proposed model, Automated Labelling using an Attention model for Radiology reports (ALARM), is only marginally inferior to an experienced neuroradiologist for normal/abnormal classication.

experiments: The experiments are impressive. The data set is large, comprised of 3000 (randomly selected out of 126,556) radiology reports produced by expert neuroradiologists consisting of all adult (> 18 years old) MRI head examinations performed between 2008 and 2019 with 5-10 sentences of image interpretation. The 3000 reports were labelled by a team of neuroradiologists to generate reference standard labels. 2000 reports were independently labelled by two
neuroradiologists for the presence or absence of any abnormality. On this coarse dataset the performance is excellent.
Another sub/classification different disease groups is made and here the performance is also good.

significance: The clinical problem that the paper addresses is very important and many research units would have a direct use of this tool in order to extract clinical data for training or research purposes that is either normal or abnormal or in given sub categories.

**Strengths:**

- the importance of the application

- the way the authors modify and re-tune the state-of-the-art BioBERT language model

- the size of the dataset that was compiled

- the performance on the normal vs abnormal task as well as the subcategory task


**Weaknesses:**

- The use of a comparison of an experienced neurologist and stroke physician vs. a neuroradiologist is somewhat strange to me; clinically speaking, either I have access to radiologists/neuroradiologists that can describe my scan or I do not. In the hospital setting, even in research, one wouldn't give scans to neurologists to describe them.

- One caveat in the experiments is though that as the results a single run on the test set is given. This is common in deep learning applications, btu makes it slightly hard to guess how this would perform if trained slightly differently.

**Justification Of Rating:**

The paper addresses a very important clinical issue and employs experiments with a considerable size data set of radiological reports of brain MRI scans that show a performance that is on par with a neuroradiologist.

**Paper Type:**

both

**Questions To Address In The Rebuttal:**

- the authors should highlight their reasoning behind the use of a comparison of an experienced neurologist and stroke physician vs. a neuroradiologist

- for people unfamiliar with NLP applications the section regarding the visualization of the word-level attention attention weights is a bit unclear. So darker color is more important, and there seems to be a lot of weights in the section regarding the spine. But why did you not exclude all the head and spine cases from the dataset as soon as you became aware of the presence of these dual examinations?

- in principle, the authors broke anonymity by including a link to the main authors GitHub repo in the article: https://github.com/tomvars/sifter/ \
While this is a matter of discussion now, I highly encourage the authors to release their code and trained network for usage and further training in other sites.

**Special Issue:**

yes

---

> ### Author Response · Authors · 2020-03-27
> **Thank you for the comments, please see our responses**
>
> .- RESPONSE 1 We thank the reviewer for their comments. Regarding the use of a neurologist, we thank the reviewer for asking for clarification here and we have made this clearer in the manuscript. For the task of coarse labelling i.e. seeing whether a report contains an abnormality or not, we could have compared our algorithm performance to the consensus of three experienced neuroradiologists alone - which is our reference standard. It would have been sufficient to confine the performance to this comparison alone. However, the large team of clinicians working on this project wanted to additionally compare to an experienced neurologist (and stroke physician) for the following reasons.
>
> 1. It is well known in medical data classification tasks that the rate limiting step is typically labelling by clinicians. We wanted to determine whether an experienced neurologist (and stroke physician) would be suitable for such a task as there are less neuroradiologists than neurologists or stroke physicians by a ratio of 1:4. To be clear, this doesn’t mean that the neurologist would assess the scan, only the report describing the scan
>
> 2. In most countries including anonymous [country], the experienced neurologist (and stroke physician) orders the scan from his/her neuroradiology colleagues, and once the scan is completed, hours-weeks later interprets the report produced by the neuroradiology colleague. In most countries including anonymous [country], neurologists (and stroke physicians) have outpatient clinics (or inpatient ward rounds) and will frequently interpret the report held on the electronic patient records during the face-to-face patient consultation. This must be the interpretation of the neurologist (and stroke physician) alone because their neuroradiology colleagues are not accessible at the time of the outpatient clinic (or inpatient ward rounds).
>
> 3. The performance of the algorithm can be put into clinical perspective. It is implicit that such an algorithm might also be a useful tool for an experienced neurologist (and stroke physician) to simply understand whether the report shows that the scan is normal or abnormal – after all, we have shown that it is challenging for such an expert neurologist (and stroke physician) to even determine whether the report shows that the scan is normal or abnormal.
>
> - Notes: To reduce the chance that the allocation of normal or abnormal by our experienced neuroradiologists somehow differed to what is considered normal or abnormal in the clinic by an experienced neurologist (and stroke physician), the neuroradiology team taught the neurologist (and stroke physician) a set of easy-to-follow rules to reduce any ambiguity, over a six month period in the run-up to their classification task (as mentioned in section 4).
>
> - Furthermore, our approach is similar to that used recently where experienced neuroradiologists were the ‘reference standard’ and comparison was made to someone who is not a neuroradiologist (e.g. bleed vs no bleed on CT head deep learning classification task – reference Kuo W,  Hӓne C,  Mukherjee P, Malik J, Yuh EL PNAS November 5, 2019 116 (45) 22737-22745.). Therefore our approach appears to be a sensible strategy.
>
> - In summary, for the task of coarse labelling i.e. seeing whether a report contains an abnormality or not, we could have compared our algorithm performance to the consensus of experienced neuroradiologists alone - which is our reference standard. However, we believe it is of vital clinical importance to also compare the performance to an experienced neurologist (and stroke physician).

---

> ### Author Response · Authors · 2020-03-27
> **Response 2**
>
> RESPONSE 2 We thank the reviewer for their comment. Regarding the point about retaining an ‘outlier’ examination, in this case a head and a spine erroneously reported together, we thank the reviewer for asking for a clearer explanation for the rationale and we have made this clearer in the manuscript.
>
> - In section 4.1 we say that “such combined reports were very rare as the number of head and spine examinations occurring at the same time was small (less than 0.5% of our data)”.
>
> - We also say “because we extracted all head examinations from a real-world hospital CRIS, occasionally the neuroradiologist who reported the original scan decided to include a spine report in the section dedicated for head reports (as opposed to a section for the spine).”
>
> - In other words the neuroradiologist would typically report heads in the section dedicated for head reports. What happened in this case is the neuroradiologist deviated from hospital protocol i.e. in a tiny fraction of this 0.5% (exact percentage from120,000 unknown).
>
> - We agree, our classification results could be improved even further by removing ALL the 0.5% of examinations where the head and spine examinations were performed concurrently, which would remove the outlier. However, our concept had always been to use ALL written head reports on an “intention-to-report” basis. Many models in deep learning suffer from domain shift when an in-sample hold out test set or, more commonly, in an out-of-sample hold out test set is tested. We therefore wanted our algorithm to be as generalisable as possible by using all real world data and had elected to use ALL head reports from a real-world hospital CRIS.

---

> ### Author Response · Authors · 2020-03-27
> **response 3**
>
> RESPONSE 3 Regarding making our code available online, we thank the reviewer for suggesting this and apologise for including a github link. This was an oversight on our part as it didn’t occur to us that this breached anonymity, but it hopefully demonstrates that we are committed to open source science. Our code, as well as the automatic labelling tool, will be made available for other researchers to label their imaging datasets upon publication.

---

### Meta-Review · Area_Chair1 · 2020-04-07
**MetaReview of Paper89 by AreaChair1**

**Rating:** 4
**Recommendation For Accepted Papers:** Best Paper Award, Oral

**Metareview:**

All reviewers recommend acceptance of the paper and the authors tried to address any remaining comments. I also think this is a topic with a lot of interest from the MIDL community.

**Paper Type:**

methodological development

**Special Issue:**

yes

---

### Decision · Program_Chairs · 2020-04-11

Accept